# Beyond Generalist LLMs: Specialist Agentic Systems for Structured Code Workflow Execution

## Abstract

The rise of Large Language Models (LLMs) has accelerated the adoption of software development agents, now commonly found as IDE extensions and standalone applications. These agents enable users with minimal programming experience to build complete applications in minutes. Typically designed as generalists, they leverage the broad capabilities of LLMs to perform a wide range of tasks. This versatility raises a key question: do specialist agents offer meaningful advantages over generalist ones, particularly given the additional development effort they require? To explore this question empirically, we focus on business process automation specifically, the transformation of tasks defined in Business Process Model and Notation (BPMN) diagrams into executable agentic workflows. We introduce a specialist workflow tailored for this purpose and evaluate its performance against generalist solutions. Our findings show that, in this context, the specialist agentic solution produces agents that outperform those generated by generalist agents such as Roo and Cline by 2.75% in accurate task completion, while reducing the token cost of agent generation by 96%. Additionally, we identify several limitations in generalist agents, including inconsistent code generation in terms of both functionality and quality. These inconsistencies hinder their applicability in industrial settings, where reliability and maintainability are critical for large-scale adoption.

## 1 Introduction

The emergence of LLMs has accelerated the rise of autonomous software agents Ferrag et al. (2025). These AI-driven agents now appear in many forms including as IDE extensions and stand-alone no-code assistants. This has made it possible for even non-programmers to build simple applications within minutes He et al. (2025). They typically function as generalists, leveraging the vast knowledge of foundational LLMs to perform a wide array of tasks. For example, an open-source coding assistant like Roo Code can plan, write, and debug code across diverse domains directly in a developer's editor Sapkota et al. (2025). Similarly, multi-agent frameworks such as FLOW, AFLOW, AutoGen, or MetaGPT coordinate several LLM agents with predefined roles (manager, coder, etc.) to tackle complex problems in a general way Niu et al. (2025); Zhang et al. (2025b); Wu et al. (2023); Hong et al. (2024). The success of these systems highlights the versatility of general-purpose LLM agents: they have been applied to everything from web browsing and data analysis to game design and UI creation Fourney et al. (2024). Works thus far have largely considered what we define as "generalist systems", a system of agents that are able to complete a wide range of tasks, with architectures suited to free exploration of various ideas to complete a task in an unspecified manner Sapkota et al. (2026). This adaptability has led to significant adoption of these tools including the popularity of several extensions that offer this feature directly in a user's Integrated Development Environment (IDE) such as Roo and Cline Sapkota et al. (2025); Cline (2025) These systems have proven beneficial as they are able to create a complete system from a single prompt with minimal user intervention. These capabilities allow users with minimal technical knowledge to create a fully functioning system from a simple idea Sapkota et al. (2026).

However, these systems have their drawbacks. Their generalist design demands extensive planning and increases token and cost overhead, which can become significant during rapid or large-scale de-

velopment. In addition, they often lack awareness of company-specific best practices, such as style guides or preferred methods. While experienced users can define constraints and refine outputs, this is an imperfect solution that does not guarantee consistent output over many generations (as discussed in Section 2). This inconsistent output results in increased technology debt that complicates future updates and maintenance.

An alternative paradigm to generalist agentic systems is what we define as a "specialist system". These systems are characterised by a well-defined and constrained agentic workflow, designed with a focus on executing a specific type of task. Rather than relying on an LLM to generate solutions from scratch, specialist systems leverage the expertise of technical subject matter experts to construct a templated workflow. Within this scaffold, the LLM performs localised reasoning making small, context-sensitive adjustments to predefined components to complete the task rather than completing it from scratch.

These specialist systems typically require a greater upfront time investment, as the workflow must first be manually designed and validated. This makes them less suitable for ephemeral use cases such as demos or proof-of-concept experiments. However, their value becomes apparent in scenarios involving large-scale deployment. When invoked repeatedly, these systems produce consistent outputs, reducing technical debt, and simplifying both debugging and integration with external systems. Moreover, the structured nature of these systems enables more effective context management. This means that developers can tightly control the information exposed to the LLM, reducing unnecessary token usage and improving performance at scale.

Recent studies have shown that while LLMs are increasingly equipped with extended context windows, their utilisation of this capacity remains uneven An et al. (2025). Furthermore, empirical evidence suggests that performance tends to degrade as more of the context window is consumed Modarressi et al. (2025). To address this, we propose a context management strategy that dynamically restricts the active context to the minimal information required for a given subtask. This strategy presents challenges in generalist systems as it requires customized prompting that is dependent on knowledge of what information is required for each subtask when constructing the workflow. This work is also highly specific for certain tasks and does not necessarily translate directly to other tasks, making it a challenging approach to take in generalist systems. This approach offers other benefits by reducing redundancy and improving model performance relative to systems that indiscriminately retain excess context (see Section 3.5 for quantitative comparisons). Given that business processes can scale well beyond the complexity of our benchmark workflows, effective context management is a critical component for maintaining performance in large-scale deployments.

Additionally, efficient context management also offers potential cost benefits. This is especially vital in business environments where standard operating procedures may evolve frequently during development and post–deployment phases and agentic workflow generation tools are often executed repeatedly for the same task. Under such conditions, reducing the volume of tokens processed through improved context management can yield substantial savings. While the cost reduction per instance may appear marginal, the cumulative impact at scale becomes significant Mei et al. (2025). As such, there is potential value in introducing specialist workflows that use manually constructed context management. This approach minimises the information exposed to the LLM in a targeted way, helping to maintain performance while enabling cost reductions when deploying these systems at scale.

## 1.1 BUSINESS PROCESSES

To evaluate the capabilities of our LLM-based agentic systems, we adopt Business Process Model and Notation (BPMN) as the structural foundation for workflow representation. Given BPMN's ubiquity in enterprise modelling and the low barrier to entry for non-technical users to author process models in this format, it serves as a practical interface for defining and orchestrating workflows (Köpke & Safan, 2024; Nour Eldin et al., 2025; Toxtli & Li, 2025; Berti et al., 2024).

Our approach focuses on transforming BPMN-defined workflows into fully operational agentic pipelines. This includes generating all necessary wrappers for deployment as FastAPI services with callable endpoints. Unlike traditional execution engines, our system enables dynamic, context-sensitive behaviour by integrating LLM-driven decision-making into otherwise static process models.

We assess the effectiveness of this transformation by examining the fidelity and adaptability of the resulting pipelines under runtime variability and partial observability (Di Ciccio et al., 2015).

## 2 METHODOLOGY

To evaluate the potential efficiency and performance gains achievable through a specialist agent, we designed one specifically for the task of converting workflows specified in BPMN into ReAcT agents. We assessed all systems using metrics related both to the agent generation process and to the individual performance of each generated agent. This dual evaluation enables us to compare the efficiency of different systems and assess the quality of their outputs.

### 2.1 WORKFLOW SELECTION

We used two deterministic workflows in our approach. One is a synthetic disengagement workflow consisting of 12 nodes and 12 edges with several branching paths for evaluation. The second one is a hypothetical fraud optimisation workflow consisting of 11 nodes and 11 edges. Both workflows are shown in Figures 2 and 3 in Section A. Each system was tested on a total of 10 successfully generated agents for each of these workflows, with each workflow being evaluated on a total of 64 test cases producing a total of 1280 test cases to evaluate agents generated by the different systems.

The selection of these relatively simple and deterministic workflows was for two reasons. Firstly, their simplicity ensures that the entire workflows can be reliably processed by the language model. This is important for assessing the system's ability to convert a workflow into an agentic system. Normally, larger and more complex workflows risk exceeding the model's context window or reasoning capacity, resulting into errors. Secondly, the chosen workflow's deterministic nature (each node's control flow is governed by predefined labels) allows exhaustive testing of all possible execution paths. We can generate a dataset covering every path the graph can take, yielding thorough coverage in evaluation. Moreover, because the workflow's behaviour is fully specified, we can easily determine ground-truth outcomes for each test case. This enables an objective measure of task completion and process adherence where we can directly verify whether the agent-produced outputs match the expected results for each path, rather than relying on LLM-based judgements. Overall, this carefully chosen workflow provides a clear, objective testbed to validate the core capabilities of our approach.

### 2.2 AGENTIC SYSTEM DESIGN

Although the above workflows could be executed with fixed Directed Acyclic Graph (DAG) frameworks (e.g., LangGraph), we opted to evaluate our system's ability to construct a ReAct agent-based solution (Yao et al., 2023b). A ReAct agent plans a sequence of actions and dynamically calls tools in an interactive loop, rather than following a strictly predefined path like in DAGs . We chose this agentic approach for several reasons:

1. **Generality and Scaling:** ReAct agents can generalize and scale to a broader range of tasks and environments. In practical settings such as customer service chatbots, workflows often require extracting information from user messages and performing multiple tool-based actions; these scenarios benefit from the flexibility of an agent that can reason and act beyond a rigid script Leocádio et al. (2024).

2. **Flexible execution:** ReAct agents offer greater adaptability during execution. For instance, a user might correct or update an earlier input or might have already provided some required information in a previous turn. A ReAct agent is likely to re-plan on the fly and handle such deviations well compared to a static DAG workflow especially if complex resets are required.

3. **Ease of authoring:** The logic of a ReAct agent is described in natural language (as prompt instructions), making it relatively easier for non-technical subject matter experts to understand and modify the workflow. Minor changes to the workflow's structure or behaviour can be made by editing prompt text, rather than altering code or diagrammatic representations. This property can significantly speed up iteration and development by domain experts.

## 2.3 PROPOSED SYSTEM

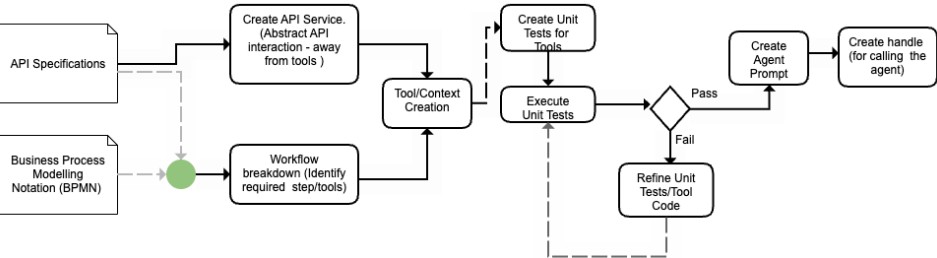

Figure 1: Proposed System Architecture

Our proposed system as in Figure 1 takes a BPMN-defined workflow and API specifications and automatically generates a working ReAct-style agent by decomposing the problem into a sequence of structured steps. In summary, the system performs the following stages:

1. **Workflow Parsing:** The BPMN workflow diagram is parsed into a sequence of discrete steps, identifying each task, decision node, and the required tool or API call at that step. This yields a structured representation of the workflow logic (including branches and conditions) that the language model can reason about.

2. **API Service Generation:** Based on a provided API specification (services or data sources the workflow needs to call), the system generates an API client service. This is essentially boilerplate code wrapping the external API calls, abstracting away low-level details. By creating a reusable API module, the complexity of tool implementation is reduced, and code duplication is minimized.

3. **Tool/Context Creation and Unit Tests:** Using the parsed workflow steps and the API service, the system then generates the code for each tool corresponding to a node or function in the workflow along with unit tests for those tools. The tests are designed to be validators of the tool's behaviour by using the deterministic nature of the workflow to assert correct outputs for given inputs.

4. **Iterative Refinement (Agent Self-Verification):** The generated tool code is executed against its unit tests. If any test fails, the system enters a refinement loop. Here, the language model analyses the failures and refines the tool implementations (and potentially the tests) to fix bugs or inaccuracies. This code-generation-and-testing loop repeats until all unit tests pass, ensuring that each component of the workflow behaves as expected in isolation.

5. **Agent Assembly and Deployment:** Once all tools are verified, the system constructs the final agent. It composes a natural language prompt that encapsulates the workflow logic (the "plan" instructions for the ReAct agent, including how to use the tools) and then generates a main function that instantiates the agent and connects it to a simple FastAPI web service. The FastAPI wrapper allows the agent to be queried in a deployed setting (e.g., simulating a user query through an API call). At this stage, the agentic system comprising of prompts, tools, and API endpoints is fully developed and ready for end-to-end testing.

## 3 EVALUATIONS AND RESULTS

### 3.1 BASELINE SYSTEMS AND EXPERIMENTAL SETUP

We evaluated the performance of our system against two recently developed automated coding agents, **Roo** and **Cline**, which are provided as Visual Studio Code extensions Microsoft. These are AI-driven tools that autonomously generate code based on instructions, like our approach. All systems were given the same BPMN workflow and API specifications. We instructed Roo and Cline to proceed with their default approach, allowing them to make their own architectural and coding decisions without additional constraints. Preliminary experiments showed that enforcing additional

design constraints often degraded output quality; we therefore allowed the baselines to operate unconstrained.

After each system declared completion with all internal tests passed and an agent produced, we conducted an end-to-end verification by invoking the agent on a set of sample inputs covering the workflow's branches. If a system's agent failed at this stage (e.g., an exception when running the full workflow), we fed the error message back into the system multiple times, allowing it to attempt automatic repairs through successive iterations until the issue was resolved or the system was no longer able to make progress in debugging. Our system's agent passed all end-to-end tests on the first attempt without any manual intervention. In contrast, the Roo and Cline agents each required one additional refinement cycles with the error feedback to achieve a working state. This difference underscored the robustness of our approach in generating correct-by-construction agents.

We also explored other automation frameworks, such as Autogen, MetaGPT and FLOW Wu et al. (2024); Hong et al. (2024); Niu et al. (2025) but these were unable to produce a functional solution for the given workflow after multiple attempts. Consequently, our comparative evaluation is focused on the Roo and Cline baselines, which were the only ones besides our own system to successfully complete the task.

## 3.2 EVALUATION METRICS

We evaluate each system across two tasks:

1. the agent generation process, and

2. the functional performance of the generated agents.

For each system under comparison, we generated 10 agents that successfully compiled and executed according to the BPMN-defined specifications. Each of the 10 agents was then evaluated on a dataset comprising all 64 possible combinations of control-flow flags, yielding 640 test cases per workflow for a total of 1280 test cases for each generation system. As mentioned in Section 2.1, the first workflow, is a synthetic disengagement workflow consisting of 12 nodes and 12 edges with several branching paths. The second one is a hypothetical fraud optimisation workflow consisting of 11 nodes and 11 edges.

## 3.3 AGENT GENERATION EVALUATION

The agent generation process was assessed using three primary metrics: task completion rate and efficiency. Efficiency covers repair iterations, and token usage. We also examined the unit test coverage to determine how well they were developed.

1. **Task Completion Rate:** This metric quantifies the number of attempts required to produce 10 fully functional agents based off the BPMNs. For Roo and Cline, we allowed the systems to iteratively refine their outputs by feeding back error traces from failed unit tests and runtime exceptions. An agent generation attempt was considered failed if the system was unable to recover from an error or deviated significantly from the intended architecture by e.g. , hardcoding logic instead of constructing an agentic system.

2. **Repair Iterations:** We measured the number of refinement loops required to produce a valid agent. For our system, this corresponds to the number of tool refinement cycles needed to pass all unit tests. For Roo and Cline, this includes both unit test failures and errors in the fastAPI service generated to invoke the agent. Notably, our systems templated approach to developing a fastAPI service to wrap the agent resulted in no errors under this category. This was reported as the average number of repair iterations necessary to complete a single successful generation. Any repairs performed in generations where the task was not successfully completed were not included.

3. **Token Usage and Cost:** We tracked the total number of tokens consumed from the initial invocation to the successful generation of a complete agent. Our system's token usage was measured using Langfuse tracing, while Roo and Cline reported their own token consumption.

| | Task Completion Rate (%) (higher is better) | Repair Iterations (avg num required per generation) (lower is better) | Token Usage (Input, 1000s) (lower is better) | Token Usage (Output, 1000s) (lower is better) | Unit Tests with inadequate coverage (avg per workflow) (lower is better) |
|---|---|---|---|---|---|
| Specialist System | **100.00**±0.00 | **0.35**±0.15 | **21.73**±1.42 | **10.11**±0.87 | **1.10**±0.35 |
| Roo | 74.07±8.59 | 1.00±0.17 | 553.26±38.99 | 15.73±0.56 | 4.70±0.57 |
| Cline | 86.96±7.77 | 0.70±0.17 | 597.23±53.30 | 14.40±0.79 | 4.95±0.76 |

Table 1: Results of Agent Generation Evaluation

4. **Unit Test Coverage:** We examined the unit tests generated for each tool by the system. Unit tests were evaluated for if they adequately covered all possible cases for a given test. This metric was reported as the number of tools generated that had inadequate coverage in some capacity, as opposed to the specific number of edge cases not being covered.

## 3.4 Agent Run Evaluations

During each of the agent runs we evaluated them on the following metrics to determine if they were able to complete their task. Each agent was run using the GPT 4.1 model to ensure consistency of results.

**Result Correctness:** Result correctness was calculated using a set of ground truth results to evaluate if the agent had successfully followed the workflow and obtained the correct answer. Given the deterministic nature of our workflows, we were able to generate ground truth results by artificially generating an input dataset that would explore all branches of the workflows and then used the workflows themselves to compute the expected outcome for each of these inputs. The tool calls necessary to complete the task was also determined based on the steps that should be taken to reach the correct end state for each piece of input data.

**Tool Correctness:** One behaviour we observed in testing was agents generating a tool that returned all necessary data, in combination with other tools that only returned a specific piece of information. This led to scenarios where despite instructions calling for multiple tools to be run, the agent obtained all necessary information to complete the task after running a single tool and computed a final answer. While this is a deviation from the instructions given to the agent, an argument can be made that the agent used its reasoning ability to determine that enough information had already been retrieved to complete the task. To report on both behaviours, we chose to adopt two separate measures to assess the tool correctness of an agent run. The first metric, "tool correctness" measures which tools they called exactly against those stated in the process. This metric was reported as tools that were not called that were expected to be called, reported as missed tool calls, as well as tool calls that were not necessary to complete the workflow, reported as excess tool calls. The second metric "information correctness" measures assess if the agent called enough tools to collect all the necessary information before returning a final answer.

By evaluating these agent run metrics we can determine which approach strikes a more appropriate balance of cost and efficiency with general performance.

## 3.5 Results

### 3.5.1 Agent Generation Results

**Task Completion Rate:** Results in Tables 1 and 3 show that our specialist system achieved full task completion across both workflows, successfully generating all 20 agents without encountering any unrecoverable errors. In contrast, the generalist systems Roo and Cline exhibited reduced reliability, failing to produce valid agents in 7 and 3 cases respectively. These failures were attributed to either persistent code errors or architectural deviations (e.g., hardcoded logic replacing agentic behaviour). Error distribution was approximately uniform across the two workflows for both generalist systems, suggesting that no workflow disproportionately contributed to failure. While these results favour the specialist system, it is important to acknowledge that performance may vary with more complex or domain-specific workflows, which could either exacerbate or mitigate error rates in generalist systems.

**Repair Iterations:** As mentioned in Section 3.3, a repair iteration is a single LLM generation cycle used to update code following a failed test or runtime error. Our specialist system required an average of 0.35 repair iterations per successful agent generation, indicating high reliability and minimal need for post-generation correction. Roo and Cline required 1.0 and 0.7 iterations respectively, reflecting a greater reliance on refinement loops to achieve functional outputs. These results further underscore the robustness of the specialist system in producing agents.

**Token Usage and Cost:** Token consumption was measured from initial invocation to successful agent generation, capturing both input and output tokens. Our specialist system exhibited significantly lower token usage, averaging 21.73k input and 10.11k output tokens per agent. In contrast, Roo consumed 553.26k input and 15.73k output tokens, while Cline required 597.23k input and 14.40k output tokens. These figures represent a 2446%/55.58% increase for Roo and a 2648%/42.43% increase for Cline relative to the specialist system.

This disparity highlights the efficiency advantages of a specialist approach, particularly in scenarios requiring large-scale deployment. While generalist systems may be suitable for ad hoc or low-frequency tasks, their elevated token consumption introduces substantial cost overhead when scaled. The results suggest that specialist systems offer a more sustainable solution for high-volume agent generation.

Beyond the specialist–generalist agentic workflow divide, we observe notable differences between Roo and Cline. Roo demonstrated marginally better token efficiency but suffered from lower task completion and higher repair iteration counts. Cline, while more costly in terms of input tokens, achieved higher success rates and required fewer refinements. Interestingly, despite Roo's greater need for repair iterations, its overall token usage remained lower suggesting that its base generation process may be more concise. These findings imply a trade-off between efficiency and reliability that should be considered when selecting generalist systems for specific use cases.

**Unit Test Coverage:** None of the evaluated systems achieved complete unit test coverage. On average, our specialist system produced 1.1 tools per agent with inadequate test coverage, compared to 4.7 and 4.95 tools for Roo and Cline respectively. While these gaps are non-trivial, poor test coverage can lead to unpredictable behaviour in deployment. For the specialist system, an LLM-as-a-judge feedback loop could be integrated to automatically assess and improve test coverage. Roo and Cline could similarly benefit from either manual review or automated refinement mechanisms. However, such enhancements would introduce additional cost and complexity, potentially offsetting the efficiency gains of the underlying systems.

### 3.5.2 AGENT RUN EVALUATION RESULTS

| | Disengagement Process: Workflow 1 | | | | Fraud Optimisation: Workflow 2 | | | |
|---|---|---|---|---|---|---|---|---|
| | **Task Completion (higher is better)** | **Missed tool calls per run (lower is better)** | **Excess tool calls per run (lower is better)** | **Information Correctness (higher is better)** | **Task Completion (higher is better)** | **Missed tool calls per run (lower is better)** | **Excess tool calls per run (lower is better)** | **Information Correctness (higher is better)** |
| **Specialist System** | **98.44**±0.49 | **0.09**±0.02 | **0.09**±0.01 | **100.00**±0.02 | **99.53**±0.27 | **0.92**±0.01 | 0.11±0.01 | **100.00**±0.02 |
| **Roo** | 98.12±0.54 | 0.98±0.04 | 0.15±0.02 | 99.53±0.02 | 94.38±0.91 | 2.01±0.03 | **0.10**±0.01 | 99.53±0.02 |
| **Cline** | 94.53±0.90 | 0.71±0.04 | 0.29±0.02 | **100.00**±0.01 | 97.97±0.56 | 2.11±0.02 | 0.16±0.02 | 99.84±0.04 |

Table 2: Overall Results of Agent Run Evaluation

All evaluated agents in Table 2 demonstrated high task completion accuracy, with success rates exceeding 90% across the board. Our specialist system achieved the highest performance, consistently surpassing 98% on both workflows. Notably, the performance gap between the generalist systems Roo and Cline was not uniform: Roo performed better on Workflow 1, whereas Cline showed stronger results on Workflow 2. This variation suggests that architectural and design choices inherent to each system may be better suited to specific workflow structures, highlighting the importance of workflow characteristics when selecting an agentic framework.

Regarding tool correctness, agents in nearly all cases successfully retrieved the information needed to complete their tasks, indicating minimal reliance on hallucinated or fabricated content across all the three systems. However, the presence of correct information does not guarantee its correct utilisation. This distinction is evident in the observed disparity between information correctness

and overall task completion, suggesting that some agents failed to apply retrieved data appropriately within the workflow logic.

Further analysis revealed that agents generated by Roo and Cline exhibited a higher tendency to omit tool calls. These agents were more likely to produce final outputs immediately after acquiring the necessary information, rather than executing the full sequence of prescribed actions. This behaviour reflects a preference for flexible task execution over strict adherence to workflow instructions. Such flexibility is not inherently undesirable and may be appropriate depending on user expectations and particularly in contexts where the ReAct framework is intended to support adaptive behaviour. These missed tool calls could likely be mitigated through prompt refinement, which may improve instruction-following and yield results more consistent with those observed in our specialist system. The specialist agents demonstrated a higher likelihood of executing all required steps faithfully. Across all three systems, we observed minimal instances of redundant tool calls that were not explicitly required in the instructions.

## 4 LITERATURE REVIEW

**BPMN and Traditional Workflow Execution:** Business Process Model and Notation (BPMN) has long served as the standard for modelling structured business workflows. Its graphical notation is both human-readable and machine-executable, enabling organisations to define, automate, and monitor processes effectively (White, 2004; Chinosi & Trombetta, 2012; Dumas et al., 2018). Traditional BPMN engines operate on deterministic, rule-based paradigms involving human tasks, service calls, and decision gateways (Weske, 2019). While robust for predictable scenarios, these engines lack the flexibility required for dynamic, context-sensitive decision-making (Van Der Aalst et al., 2020). Early efforts to enhance BPM systems introduced agent-based automation (Wooldridge & Jennings, 1995), with BPMN providing a consistent modelling language to support this evolution (Rosemann & vom Brocke, 2014). Extensions to BPMN have been proposed to accommodate more complex and adaptive workflows (Braun et al., 2014), yet execution remains constrained by static semantics (Mendling et al., 2018). Adaptive workflow systems (Reichert & Weber, 2012), context-aware frameworks (Rosemann et al., 2008), and decision-centric models (Batoulis et al., 2015) have attempted to address these limitations. Nonetheless, traditional BPMN execution continues to struggle with unstructured data, runtime variability, and ambiguous decision logic (Marrella, 2019).

**LLMs and Agentic Workflows:** Recent advances in Large Language Models (LLMs) have shifted their role from passive predictors to interactive agents capable of reasoning, planning, and executing tasks from natural language instructions (Wei et al., 2022). Agentic workflows leverage this capability to interleave reasoning with tool use, enabling autonomous task completion with minimal human oversight (Schick et al., 2023). Architectures such as ReAct (Yao et al., 2023b) and Tree-of-Thoughts (Yao et al., 2023a) exemplify this shift, combining structured reasoning with action execution. For tasks requiring symbolic precision, approaches like PAL externalise reasoning through program synthesis (Gao et al., 2023). Agentic workflows represent a departure from traditional automation, enabling systems to dynamically coordinate tasks and make decisions (Yang et al., 2023). Research has explored agent architectures with external memory, planning modules, and tool integration (Wu et al., 2024). AFLOW (Zhang et al., 2025b) reformulates workflow optimisation as a code search problem, using Monte Carlo Tree Search to refine LLM-driven workflows. MaAS (Zhang et al., 2025a) introduces a probabilistic multi-agent framework, sampling task-specific agent teams from a supernet to adaptively construct workflows across domains. These developments demonstrate the potential of agentic AI for complex tasks such as retrieval, analysis, and decision-making (Singhal et al., 2023). However, they also introduce challenges in traceability, control, and integration with structured systems like BPMN (Mialon et al., 2023; Deng et al., 2023).

**BPMN and LLM Integration:** Initial work at the intersection of BPMN and LLMs has focused primarily on modelling rather than execution. Techniques include generating BPMN diagrams from textual descriptions, conversational refinement of models, and workflow mining (Köpke & Safan, 2024; Nour Eldin et al., 2025; Toxtli & Li, 2025; Berti et al., 2024). These approaches improve accessibility for non-experts but often treat BPMN as a static artefact.

Recent efforts have shifted towards agentic automation, where LLMs synthesise workflows and execute tasks across tools and APIs (Jain et al., 2024; Zeng et al., 2023; Ye et al., 2023). Despite progress, a key gap remains: bridging BPMN's formal structure with the flexibility of agentic sys-

tems. Enabling LLM agents to reliably execute BPMN-defined workflows while preserving semantic rigour and adapting to unstructured inputs remains an open challenge.

## 5 CONCLUSION

We present a specialist agentic system for transforming BPMN-defined workflows into executable ReAct agents, demonstrating improved reliability and efficiency over generalist agentic solutions. Our system achieves higher task completion rates and significantly lower token consumption, underscoring the benefits of structured agent generation in business process automation. By decomposing workflows into modular components and integrating iterative refinement, the system yields robust, testable agents that generalise across a range of workflow configurations. While our evaluation focuses on deterministic workflows of moderate complexity, future work will address scalability to larger, more intricate workflows featuring deeper branching and non-deterministic elements. This direction aims to broaden the applicability of specialist agentic systems to complex enterprise scenarios, where adaptability and performance guarantees are essential.

## 6 GENAI USAGE DISCLOSURE

In line with our experimental design, LLMs and LLM-powered systems were used to generate and execute the agentic workflows evaluated in this paper. We also used LLMs to enhance our text through light editing tasks such as grammar correction, word autocorrection, and sentence restructuring.

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

## A  APPENDIX

Pseudocode 1 outlines the decision logic used to classify as set of users as either "engaged" or "disengaged". As indicated in Figure 2, this is based on proprietary eligibility, status, and other indicators. This workflow was used to label user trajectories and assess agentic system performance in downstream tasks.

**Workflow 1:**

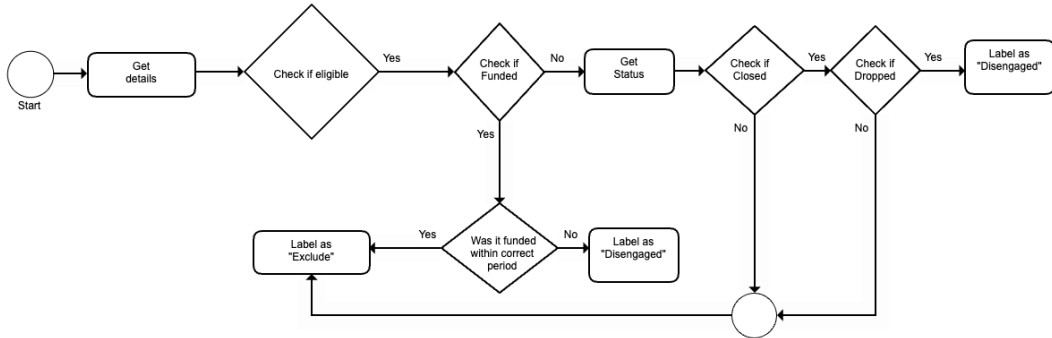

Figure 2: Disengagement Process Workflow

**Workflow 2** as in Figure 3 focuses on optimising a hypothetical fraud detection ruleset for transactions. The pseudocode in 2 captures the decision-making process for classifying transactions based on rule evaluations, operational costs, and customer experience considerations. This workflow was used to label transaction outcomes and evaluate the agentic system performance in handling complex decision logic.

**Workflow 2:**

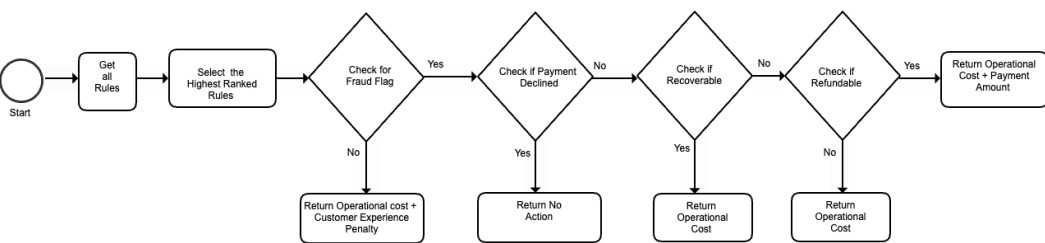

Figure 3: Fraud Optimisation Workflow

**Algorithm 1:** User labeling workflow for disengagement detection. For each user: (1) if not eligible, label as "Exclude"; else (2) if funded, check funded date - if within the target period label "Exclude", otherwise "Disengaged"; else (3) fetch status - if status is "Closed" and application is "Dropped", label "Disengaged", otherwise "Exclude".

---

**Input:** User list
**Output:** Each user labelled as "Disengaged" or "Exclude"
**for** *each user* **do**
    Get details;
    // Step 1: Eligibility check
    **if** *user is not eligible* **then**
        Label ← "Exclude";
    **end**
    **else**
        // Step 2: Funded check
        **if** *user is funded* **then**
            // Step 2a: Funded date check
            **if** *funded within the correct period* **then**
                Label ← "Exclude";
            **end**
            **else**
                Label ← "Disengaged";
            **end**
        **end**
        **else**
            // Step 3: Status check
            Get Status;
            **if** *status is "Closed"* **then**
                // Step 3a: Dropped check
                **if** *application is "Dropped"* **then**
                    Label ← "Disengaged";
                **end**
                **else**
                  Label ← "Exclude";
                **end**
            **end**
            **else**
                Label ← "Exclude";
            **end**
        **end**
    **end**
**end**

---

**Algorithm 2:** Decision workflow to compute the return value based on fraud status, payment outcome, and recoverability/refundability. The algorithm first gathers all rules and selects the highest-ranked ones, then returns: (1) Operational Cost + Customer Experience Penalty if no fraud flag is set; else (2) No Action if the payment is declined; else (3) Operational Cost if the case is recoverable; else (4) Operational Cost + Payment Amount if refundable; otherwise (5) Operational Cost.

**Input:** Rule list
**Output:** Return value based on conditions
Get all Rules;
Select the highest ranked Rules;
// Step 1:  Fraud flag check
**if** *Fraud flag is NOT set* **then**
  | Return Operational Cost + Customer Experience Penalty;
**end**
**else**
  | // Step 2:  Payment declined check
  | **if** *Payment is Declined* **then**
  |   | Return No Action;
  | **end**
  | **else**
  |   | // Step 3:  Recoverable check
  |   | **if** *Recoverable* **then**
  |   |   | Return Operational Cost;
  |   | **end**
  |   | **else**
  |   |   | // Step 4:  Refundable check
  |   |   | **if** *Refundable* **then**
  |   |   |   | Return Operational Cost + Payment Amount;
  |   |   | **end**
  |   |   | **else**
  |   |   |   | Return Operational Cost;
  |   |   | **end**
  |   | **end**
  | **end**
**end**

| | Disengagement Process: Workflow 1 | | | | | Fraud Optimisation: Workflow 2 | | | | |
|---|---|---|---|---|---|---|---|---|---|---|
| | Task Comp. Rate (%) | Input Tokens (1000s) | Output Tokens (1000s) | Repair Iterations | Unit Test Coverage (tools with inadequate coverage per generation) | Task Comp. Rate (%) | Input Tokens (1000s) | Output Tokens (1000s) | Repair Iterations | Unit Test Coverage (tools with inadequate coverage per generation) |
| Specialist Sytem | **100.00**±0.00 | **23.78**±2.43 | **12.55**±1.31 | **0.70**±0.26 | **0.90**±0.18 | **100.00**±0 | **19.68**±1.3 | **7.67**±0.39 | **0.00**±0.00 | **1.30**±0.70 |
| Roo | 76.92±12.16 | 568.59±58.56 | 13.86±0.78 | 0.90±0.28 | 3.40±0.46 | 71.42±12.53 | 536.13±53.99 | 17.60±1.01 | 1.10±0.23 | 6.00±1.11 |
| Cline | 83.33±11.24 | 507.10±63.01 | 12.65±1.13 | **0.70**±0.22 | 4.90±1.19 | 90.91±9.09 | 687.36±82.74 | 16.14±0.79 | 0.70±0.26 | 5.00±0.95 |

Table 3: Agent Generation Evaluation Metrics Per Workflow

