# OpenReview forum: "Beyond Generalist LLMs: Specialist Agentic Systems For Structured Code Workflow Execution"
_ICLR.cc/2026/Conference — ICLR 2026 Conference Withdrawn Submission_

### Official Review · Reviewer_tWsF · 2025-10-29

**Soundness:** 2
**Presentation:** 3
**Contribution:** 2
**Rating:** 2
**Confidence:** 3

**Summary:**

This paper proposes a specialist agentic system for converting BPMN-defined workflows into executable ReAct agents. The authors argue that such a specialized approach outperforms generalist LLM-based coding agents (e.g., Roo, Cline) in terms of task completion accuracy and token efficiency. The system decomposes the workflow generation process into structured stages: parsing, API generation, tool creation, iterative refinement, and deployment. Evaluation is conducted on two deterministic workflows, with results indicating that the specialist system achieves higher task completion rates (≈2.75% better) and significantly lower token consumption (≈96% reduction) compared to the baselines.

**Strengths:**

1. The emphasis on dynamic context restriction is well-aligned with recent findings on LLM performance degradation with extended context, and the paper provides quantitative support for its benefits.
2. The paper addresses a relevant and practical problem—efficiency and reliability in automated workflow execution—which is of growing interest in both academic and industrial settings.

**Weaknesses:**

1. The core idea—using a structured, template-driven approach to guide LLM code generation—is not fundamentally new. Similar concepts have been explored in prompt engineering, program synthesis, and tool-assisted LLM frameworks. The paper does not sufficiently differentiate its contribution from prior work in modular or constrained code generation.
2. The selection of baselines (Roo, Cline) is reasonable, but the failure of other multi-agent frameworks (e.g., AutoGen, MetaGPT) to produce working solutions is not deeply analyzed. This limits the understanding of *why* those systems failed and whether the proposed method is truly more general or simply better tuned for this specific task.
3. The contribution of individual system components (e.g., iterative refinement, context management) is not isolated or evaluated. It is unclear which parts of the pipeline are most critical to the performance gains.

**Questions:**

1. How would your system perform on workflows with non-deterministic nodes, human-in-the-loop steps, or processes involving external APIs with unpredictable latency or failure modes?
2. What specific aspects of your system are novel compared to existing template-based or scaffolded code generation approaches? Can you cite and contrast with related work in modular LLM-based code generation?
3. Have you conducted ablation studies to determine which stages of your pipeline (e.g., parsing, self-verification, context management) contribute most to the performance improvements?
4. Your workflows are relatively small. How does the system handle larger BPMN models (e.g., 50+ nodes) or workflows that exceed the context window of the underlying LLM?

---

### Official Review · Reviewer_YDCM · 2025-10-30

**Soundness:** 1
**Presentation:** 1
**Contribution:** 2
**Rating:** 2
**Confidence:** 2

**Summary:**

This paper proposes a specialist agentic system that transforms BPMN diagrams into executable agentic workflows. It evaluates the baseline performance of generalist solutions and confirms the applicability of the proposed system under production.

**Strengths:**

The paper is written clearly and easy to follow.

**Weaknesses:**

The paper is more like an engineering work rather than an academic innovation.

**Questions:**

- The definition of the problem is not clear, and the evaluation itself does not follow serious academic protocols.
- The reviewer believes the entire system has merits specifically for the BPMN represented workflows, but the method itself does not go through extensive performance studies.
- The illustrations/tables are kind-of rough. Please consider polish them.
- Please polish the manuscript carefully, especially considering for the problem you are addressing, which/what are the fundamental problems to be solved, and what are the popular baselines to be considered. The reviewer is quite unfamiliar with the experimental settings and cannot make judgement on the technical merits of the method.

---

### Official Review · Reviewer_7VXW · 2025-10-31

**Soundness:** 3
**Presentation:** 2
**Contribution:** 2
**Rating:** 4
**Confidence:** 4

**Summary:**

This paper introduces a specialist agentic compilation framework aiming at business process automation, of which the core objective is transforming business process models and workflows defined by BPMN into executable ReAct agents. The main contributions of this paper include: 1) utilizing domain expertise to construct a structured workflow framework that integrates key modules such as workflow parsing, API service generation, and unit testing-driven iterative optimization; 2) conducting empirical comparisons against general-purpose agents Roo and Cline, demonstrating advantages in task completion rates (100%), token costs (reduced by 96%), and output consistency; 3) addressing the issues of output instability and high token overhead in general-purpose systems, and discussing the adaptability of the specialized system in large-scale industrial deployment scenarios. The experiments show that the solutions provided by specialist agents are  superior in performance and more token-friendly than those created by generalist coding assistants.

**Strengths:**

1. The framework's modular design simplifies debugging and extension, and the architecture maps cleanly onto industrial workflows.

2. An empirical comparison against general-purpose frameworks (Roo, Cline) on 1,280 test cases shows consistent gains in task-completion rate, token consumption, repair iterations, and test coverage.

3. A dynamic context-restriction strategy is proposed that selectively exposes only the information required at each step, simultaneously improving model accuracy and lowering cost.

**Weaknesses:**

1. Although the authors explains that frameworks like AutoGen, MetaGPT, and FLOW were excluded due to their “multiple attempts failing to generate functional solutions,” they do not specify the exact reasons for these failures (for example, whether they were unable to understand BPMN semantics or complete FastAPI service encapsulation). The lack of clarity may hinder the ability to accurately assess the reasons behind the failures.

2. The tests only cover two small-scale deterministic workflows (with 11-12 nodes and edges each) and do not reflect the complexity often found in enterprise-level BPMN processes, limiting the proof of the system's adaptability to real-world complex workflows. In addition, the analysis of non-deterministic workflows (e.g., those containing dynamic decision rules) are not included, raising doubts about the system's generalization capabilities.

3. The evaluation focuses on functional correctness and token cost, yet omits deployment-critical metrics such as response latency, concurrency behavior, and end-user experience.

4. The core modules (e.g., the mapping logic between BPMN nodes and ReAct actions, strategies for correcting code after testing failures) lack specific technical details. The authors also do not provide open-source code, datasets, or BPMN workflow specifications, making it difficult for other researchers to reproduce the experimental results.

**Questions:**

1.The authors mention that AutoGen and MetaGPT are excluded due to their “inability to generate functional solutions.” Can the authors elaborate on the specific scenarios where these frameworks failed? Were adjustments to prompts or input specifications tried to optimize their performance?

2.Why was LangChain Agents not tested? Was this frameworks proven to be completely incapable of supporting BPMN workflow conversion in preliminary experiments?

3.Regarding complex BPMN workflows with multiple nested sub-processes, how does the system perform in terms of parsing and conversion? Have any related tests been conducted, and can results or technical adaptation solutions be shared?

4.In the dynamic context management strategy, what criteria are used to filter “minimal necessary information” (e.g., based on BPMN node types, sub-task dependencies)? What specific algorithms or rules were implemented to achieve this filtering logic?

---

### Official Review · Reviewer_jUxm · 2025-11-01

**Soundness:** 1
**Presentation:** 2
**Contribution:** 1
**Rating:** 2
**Confidence:** 3

**Summary:**

This paper explores whether specialist agentic systems outperform generalist LLM-driven agentic systems in business process automation, focusing on transforming BPMN-defined tasks into executable agentic workflows. It proposes a specialist workflow and evaluates it against generalist baselines (Roo and Cline) using two workflows. Results show the specialist system outperforms generalists in accurate task completion, reduces token costs significantly, and achieves a higher task completion rate in agent generation.

**Strengths:**

The paper is easy to understand with a clear structure.

**Weaknesses:**

The paper's completeness is low. Experiments are not comprehensive enough. The method lacks meaningful insights to contextualize results, with no exploration of why the specialist system performs better or discussion of limitations and generalizability beyond the specific setup. Additionally, the work leans more toward an engineering implementation (focused on building and testing a specialist workflow) rather than a research paper.

**Questions:**

See weaknesses.

---

### Note · Authors · 2025-12-03

I have read and agree with the venue's withdrawal policy on behalf of myself and my co-authors.